# Pressure-Tuning Superconductivity in Noncentrosymmetric Topological Materials ZrRuAs

**DOI:** 10.3390/ma15217694

**Published:** 2022-11-01

**Authors:** Changhua Li, Yunlong Su, Cuiwei Zhang, Cuiying Pei, Weizheng Cao, Qi Wang, Yi Zhao, Lingling Gao, Shihao Zhu, Mingxin Zhang, Yulin Chen, Youguo Shi, Gang Li, Yanpeng Qi

**Affiliations:** 1School of Physical Science and Technology, ShanghaiTech University, Shanghai 201210, China; 2Beijing National Laboratory for Condensed Matter Physics, Institute of Physics, Chinese Academy of Sciences, Beijing 100190, China; 3ShanghaiTech Laboratory for Topological Physics, ShanghaiTech University, Shanghai 201210, China; 4Department of Physics, University of Oxford, Parks Road, Oxford OX1 3PU, UK; 5Shanghai Key Laboratory of High-Resolution Electron Microscopy, ShanghaiTech University, Shanghai 201210, China

**Keywords:** high pressure, superconductivity, topological materials, noncentrosymmetric

## Abstract

Recently, the hexagonal phase of ternary transition metal pnictides TT’X (T = Zr, Hf; T’ = Ru; X = P, As), which are well-known noncentrosymmetric superconductors, were predicted to host nontrivial bulk topology. In this work, we systematically investigate the electronic responses of ZrRuAs to external pressure. At ambient pressure, ZrRuAs show superconductivity with *T*_c_ ~ 7.74 K, while a large upper critical field ~ 13.03 T is obtained for ZrRuAs, which is comparable to the weak-coupling Pauli limit. The resistivity of ZrRuAs exhibits a non-monotonic evolution with increasing pressure. The superconducting transition temperature *T_c_* increases with applied pressure and reaches a maximum value of 7.93 K at 2.1 GPa, followed by a decrease. The nontrivial topology is robust and persists up to the high-pressure regime. Considering both robust superconductivity and intriguing topology in this material, our results could contribute to studies of the interplay between topological electronic states and superconductivity.

## 1. Introduction

Superconductivity in topological quantum materials is a striking phenomenon, and it constructs the pivotal pathway for searching for topological superconductors (TSC), which host a full pairing gap in the bulk, but Majorana-bound states at the surface [1,2,3,4]. The intrinsic bulk superconductivity can induce two-dimensional (2D) TSC in the presence of the topological surface Dirac-cone states. The superconducting gap of the predicted topological surface Dirac-cone states has been confirmed experimentally in FeSe-based superconductors by angle-resolved photoemission spectroscopy (ARPES) and scanning tunneling microscope (STM) measurements [5,6]. Systematic study for 3D TSC remains rare due to the lack of suitable candidates.

A feasible route to realizing topological superconductors is to search for intrinsic superconductivity in topological materials. In this work, we focus on ternary equiatomic transition metal phosphides TT’X (T = Zr, Hf; T’ = Ru; X = P, As). According to the literature [7,8,9,10], there are four different types of crystal structures for these compounds, i.e., the TiNiSi-type orthorhombic structure (o-phase), the TiFeSi-type orthorhombic structure (o’-phase), the Fe_2_P-type hexagonal structure (h-phase), and the MgZn_2_-type hexagonal structure (h’-phase). Both ZrRuAs and HfRuP belong to Fe_2_P-type structures and show intrinsic noncentrosymmetric superconductivity with relatively high transition temperatures. Recently the first-principles calculations predicted that both ZrRuAs and HfRuP host nontrivial bulk topology [8]. When ignoring the spin-orbit coupling (SOC), ZrRuAs/HfRuP possess two nodal rings slightly above the Fermi energy (*E_F_*) in the *k_z_* = 0 planes. However, when considering the SOC, they enter either a Weyl semimetal (WSM) phase (e.g., HfRuP) due to the lack of inversion symmetry or a topological crystalline insulating (TCI) phase (e.g., ZrRuAs) with trivial Fu-Kane *Z*_2_ indices but nontrivial mirror Chern numbers. Combined with noncentrosymmetric superconductivity, the nontrivial topology of the normal state in ZrRuAs/HfRuP may generate unconventional superconductivity in both the bulk and surfaces.

Pressure is an effective method to tune the lattice structure and to manipulate the electronic state, such as superconductivity and topological phases of matter, without introducing impurities [11,12,13,14,15]. Despite the superconductivity in ternary equiatomic transition metal phosphides TT’X being reported a few decades ago [10], a detailed study of its superconducting properties, in particular under high-pressure association with topological states, is still lacking. In this work, we systematically investigate the superconducting properties of ZrRuAs in both ambient pressure and high-pressure conditions. ZrRuAs with noncentrosymmetric crystals shows bulk superconductivity at 7.74 K, while a large upper critical field is obtained, which is commensurate to the weak-coupling Pauli limit. The superconducting transition temperature *T_c_* increases with applied pressure and reaches a maximum value of 7.93 K at 2.1 GPa, then decreases with a dome-like behavior. We find that the application of pressure does not qualitatively change the electronic and topological nature of the two systems until the high-pressure regime, based on our ab initio band structure calculations.

## 2. Experimental Detail

Cu-As and Cu-P fluxes were used to grow the single crystals ZrRuAs and HfRuP, respectively. High-purity Zr (Hf), Ru, As (P), and Cu elements were sealed in tantalum capsules and then in quartz tubes. The quartz tubes were first heated to 1273 K for 20 h to pre-sinter the Cu-As or Cu-P alloys. Then, the tubes were heated to 1433 K, maintained for 10 h, and then slowly cooled down to 1373 K. The quartz tubes were immediately quenched into ice water to stabilize the hexagonal crystalline structure of the ZrRuAs and HfRuP samples. After that, bar-shaped single crystals were yielded after dissolving the Cu-As or Cu-P fluxes in nitric acid. The crystal surface morphology and composition were examined by scanning electron microscopy (SEM) and energy dispersive x-ray spectroscopy (EDS) analysis (Phenom Prox). The single crystal diffraction pattern was obtained using a Bruker dual sources single crystal X-ray diffractometer at room temperature, and the X-ray source comes from a molybdenum target. The dependence of direct-current electrical resistivity (*ρ*) on temperature was measured at 1.8−300 K using a conventional four-probe method. The magnetization measurement was carried out on a Magnetic Property Measurement System (MPMS). In situ high-pressure resistivity measurements were performed in a nonmagnetic diamond anvil cell (DAC). A piece of nonmagnetic BeCu was used as the gasket. A cubic BN/epoxy mixture layer was inserted between the BeCu gasket and electrical leads as an insulator layer. Four Pt foils were arranged in a van der Pauw four-probe configuration to contact the sample in the chamber for resistivity measurements. The pressure was determined by the ruby luminescence method [16].

The ab initio calculations were performed within the framework of density functional theory (DFT) as implemented in the Vienna ab initio simulation package (VASP) with the exchange-correlation function considered in the generalized gradient approximation potential. A *k*-mesh of 9 × 9 × 15 and a total energy tolerance of 10^−5^ eV were adopted for relaxations of structures and calculations of electronic band structures. The spin-orbital coupling was considered self-consistent in this work. Wannier charge centers were calculated by *Z*2Pack [17] through tight binding models based on the maximally localized Wannier functions as obtained through the VASP2WANNIER90 interfaces in a non-self-consistent calculation.

## 3. Results and Discussion

As shown in Figure 1a, ZrRuAs crystallized in the hexagonal structure with space group $\;P\bar{6}2m$ (No. 189). This is a typical layered structure with a naturally broken space-inversion symmetry. In each layer, either Zr and As atoms or Ru and As atoms occupy the hexagonal lattice. All atoms have positions in the layer parallel to the crystallographic *ab*-plane half of the lattice constant *c*. The triangular clusters of three Ru atoms are formed in the *ab*-plane. Figure 1b shows the composition and morphology of ZrRuAs single crystal. The average compositions were derived from a typical EDS measurement at several points on the crystal, revealing good stoichiometry with the atomic ratio of Zr:Ru:As = 31.57:37.02:31.42. Both the optical microscope and scanning electron microscope (Figure 1c) show a hexagonal rod-like crystal. Figure 1d shows the single crystal X-ray diffraction pattern on (010) plane. The upper left region is replaced by a standard Fe_2_P-type structure diffraction pattern is used to prove the same crystal structure.

Prior to in situ high-pressure measurements, we performed the transport measurement for ZrRuAs at ambient pressure. Figure 2a shows the typical *ρ*(*T*) curves of ZrRuAs from 1.8 K to 300 K. Considering the bar-shaped single crystals, the temperature dependences of resistivity (*ρ_xx_*) of ZrRuAs are measured with the dc current along the *c*-axis direction. Under zero field, ZrRuAs exhibits a typical metallic behavior with residual resistivity of *ρ*_0_ = 0.125 mΩ cm. At low temperatures, a sharp drop in *ρ*(*T*) to zero was observed, suggesting the onset of a superconducting transition. The inset of Figure 2a shows an enlargement of the superconducting transition. The transition temperature *T_c_* at 7.44 K in our single crystal is lower than the previous report in polycrystalline samples. The transition widths Δ*T_c_* are approximately 0.81 K, implying fairly good sample quality. The bulk superconductivity of ZrRuAs was further confirmed by the large diamagnetic signals and the saturation trend with the decreasing temperature. As shown in Figure 2b, the rapid drop of the zero-field-cooling data denotes the onset of superconductivity, consistent with the temperature of zero resistivity. The difference between zero-field-cooling and field-cooling curves indicates the typical behavior of type-II superconductors. We conducted resistivity measurements in the vicinity of *T*_c_ for various external magnetic fields. As can be seen in Figure 2c, the resistivity drops shift to a lower temperature with an increasing magnetic field and keep a superconducting state even in the field of 9 T, indicating the high upper critical field (*H_c_*_2_). We extract the field (*H*) dependence of *T_c_* for ZrRuAs and plot the *H*(*T_c_*) in Figure 2d. We also tried to use the Ginzburg-Landau formula to fit the data [18]:(1)$$H_{c}\left(T\right)=H_{c}\left(0\right)\frac{1-{\left(\frac{T}{T_{c}}\right)}^{2}}{1+{\left(\frac{T}{T_{c}}\right)}^{2}}$$

It was found that the upper critical field *H*_c2_ = 13.03 T, which is comparable with the weak coupling Pauli limit value of 1.84 *T*_c_ = 13.69 T. According to the relationship between *H*_c2_ and the coherence length *ξ*, namely, *H_c_*_2_ *=*
*Φ*_0_/(2*πξ*^2^), where *Φ*_0_
*=* 2.07 × 10^−15^ Wb is the flux quantum, the derived *ξ_GL_*(0) was 5.03 nm.

To determine the lower critical field *H_c_*_1_, the field-dependent magnetization *M*(*H*) of ZrRuAs was measured at various temperatures up to *T*_c_ using a ZFC protocol. Some representative *M*(*H*) curves are shown in Figure 2e. For each temperature, *H_c_*_1_ was determined as the value where *M*(*H*) deviates from linearity (dotted line) and summarized in Figure 2f. A linear fit gives *μ*_0_*H_c_*_1_(0) = 51.85 ± 0.6 Oe. Using the formula [18]:(2)$$\mu_{0}H_{c1}\left(0\right)=\frac{\Phi_{0}}{4\pi \lambda^{2}}\ln(\frac{\lambda }{\xi })$$

We obtain the penetration depth *λ* = 5.034 nm. The calculated GL parameter of *κ* = *λ*/*ξ* ∼ 1 corroborates that ZrRuAs is a type-II superconductor.

It is well known that high pressure is a clean way to adjust lattice structures and the corresponding electronic states in a systematic fashion [10,19,20,21,22,23,24,25,26,27]. Hence, we measured *ρ*(*T*) for both ZrRuAs and HfRuP at various pressure. Figure 3a presents the temperature dependence of the resistivity of ZrRuAs for pressure up to 29.9 GPa. The room temperature resistivity (*ρ*_300K_) of ZrRuAs exhibits a non-monotonic evolution with increasing pressure. Over the whole temperature range, the *ρ*_300K_ is first suppressed with applied pressure and reaches a minimum value at about 17.0 GPa, then displays a moderate increase with further increasing pressure until 29.9 GPa. A similar evolution of *ρ*_300K_ is observed for HfRuP in Figure 3b; the lowest *ρ*_300K_ occur at 18.5 GPa and turn into an increase as the pressure increase until 32.3 GPa. In Figure 3c, it is found that the superconducting transition temperature (*T_c_*) of ZrRuAs increases from ∼7.74 K to a maximum of ~7.93 K at 2 GPa. Beyond this pressure, *T_c_* decreases slowly, showing a dome-like behavior. A similar evolution is observed for HfRuP, as shown in Figure 3d; a maximum *T_c_* of 6.50 K is attained at *P* = 2.8 GPa, then *T*_c_ fluctuates as the pressure increase to 9.8 GPa and eventually drops rapidly until 32.3 GPa.

Figure 4 shows the evolution of resistivity *ρ*_300K_ and *T_c_*s with the pressure of ZrRuAs and HfRuP; the two materials show the same variation trend under exerted pressure. The high *T_c_* under various pressures of HfRuP can last longer than ZrRuAs, but once the *T_c_* breaks into a fast drop, the slopes of HfRuP and ZrRuAs are similar. The *T*_c_-Pressure phase diagram indicates that the TT’X family could have the same dome-like shape. It is worth mentioning that both ZrRuAs and HfRuP show the difference of imposed pressure between corresponding the max *T_c_* and the minimum *ρ_c_*. Although there are inevitable errors in the measurement of the absolute value of resistivity root in the thickness of the gasket, the tendency of resistivity under pressure is reliable.

We theoretically studied the electronic structures and the topological properties of ZrRuAs under pressure. To get the crystal structure under experimental pressures, we started with the lattice constants of the pristine structure [28] and uniformly strained the three lattice constants until the calculated pressure agreed with the experimental ones. We fixed the crystal volume and allowed the atomic positions and lattice constants to change in the structure optimization. The relaxed crystal structures under various experimentally examined pressures are presented in Appendix A. The theoretically determined pressure in DFT is shown in the first column of Appendix A. The fractional atomic coordinates of Zr and Ru only change from 0.582 to 0.581 and 0.244 to 0.245 with the increase in pressure, respectively. Therefore, in these systems, pressures only shrink the crystal volume while keeping the atomic positions almost unchanged. The evolution of the electronic structure with the increase in pressure can be found in Appendix A. The change in the crystal volume barely changes the electronic structure. Thus, we expect the topological nature of these systems to remain unchanged.

We take ZrRuAs at 0.3 GPa as an example to examine our expectations. In Figure 5, nodal lines are present on the *k_z_* = 0 plane between *M*-*K* without SOC. Further, including SOC, the nodal lines are fully gapped, leading to a continuous gap between valance and conduction bands. As explained later, the gapping of the nodal lines gives nontrivial mirror Chern numbers on *k_z_ =* 0, *k_y_* = 0, and other equivalent mirror planes.

The space group of ZrRuAs contains four mirror symmetries, ${\hat{m}}_{z}$, ${\hat{m}}_{y}$, and another two equivalent ones related to ${\hat{m}}_{y}$ by ${\hat{C}}_{3}$. For each mirror symmetry operator $\hat{m}$, mirror symmetric planes are defined by a set of points $\left\{\vec{k}|\hat{m}\vec{k}=\vec{k}+\vec{G}\right\}$ in reciprocal space, where $\vec{G}$ is a reciprocal lattice vector. For example, for ZrRuAs, the mirror-symmetric planes are at *k_z_ =* 0, *π*, *k_y_* = 0, and the two equivalent ones relate to *k_y_* = 0 by ${\hat{C}}_{3}$. On the mirror plane, mirror operators take definite eigenvalues *+i, −i*. Because mirror operations belong to the little group, the Hamiltonian on the mirror planes becomes block diagonal with eigenvalues of mirror operators *+i*, *−i*. With time-reversal symmetry, the Chern number in any mirror plane is always zero. However, we can define a nonzero Chern number $C_{+i}$, $C_{-i}$ separately for each subspace. In the analogy of the spin Chern number, the mirror Chern number is defined as $C_{M}=\left(C_{+i}-C_{-i}\right)/2$ [29]. On mirror symmetric planes, time-reversal symmetry transforms eigenstates with opposite momentum in different subspaces into each other, which makes them a Kramer pair. Similar to the time-reversal invariant $Z_{2}$, mirror Chern numbers can be obtained by calculating the Wilson loops in half of the mirror planes. With the Wilson loop, the mirror Chern number is defined as $C_{M}=X_{+i}-X_{-i}$, where $X_{\pm i}$ are the differences in numbers of Wilson bands with positive and negative slopes crossing a horizontal reference line in half of the mirror plane in the subspace with eigenvalues ±i [8].

We note that Wilson loops on *k_z_* = 0, *π*, and *k_y_* = 0 planes are sufficient to derive the mirror Chern numbers of all mirror symmetric planes. From the analysis above, we can directly read mirror Chern numbers at 0.3 GPa from the Wilson loops colored by eigenvalues of mirror operators in Figure 6. $C_{M}$ are 2 for both *k_z_* = 0 and *k_y_* = 0 planes, and it is zero for *k_z_* = *π* plane. The mirror Chern numbers change when we increase pressures. We list the mirror Chern numbers at different pressures in Table 1, and the corresponding Wilson loops can be found in Appendix A. Table 1 shows that ZrRuAs is always a topological crystalline insulator with nontrivial mirror Chern numbers at different pressures. Therefore, we conclude that ZrRuAs remain topological in all pressures studied in our experiment. 

## 4. Conclusions

In summary, we have performed a comprehensive high-pressure study on topological crystalline insulator ZrRuAs. The superconducting transition temperature *T*_c_ increases from 7.44 K at ambient pressure and reaches a maximum value of 7.93 K at 2.1 GPa, then decreases with a dome-like behavior. The slight change of *T_c_*s of ZrRuAs indicates that the superconducting state is robust to external pressure. Through ab initio band structure calculations, we find that the application of pressure does not qualitatively change the electronic and topological nature of ZrRuAs until the high-pressure regime. Based on the observation that both topological properties and superconducting state are robust in ZrRuAs under high pressure, our research could not only intrigue studies of the interplay between topological electronic states and superconductivity but also inspire the experimental searching for possible topological superconductivity.

## Figures and Tables

**Figure 1 materials-15-07694-f001:**
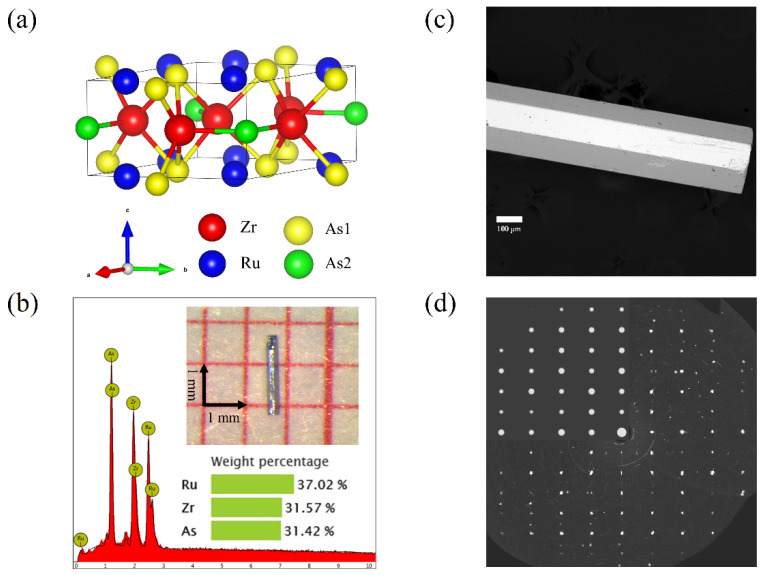
(**a**) Crystal structure of TT’X (take ZrRuAs as an example). T and T^’^ are represented by red and blue spheres, respectively, and the two types of X environment are indicated by the yellow and green spheres. The unit cell is shown by the parallelepiped. (**b**) EDS and optical photograph of ZrRuAs. (**c**) SEM images of ZrRuAs. (**d**) Single crystal diffraction pattern of Fe_2_P-type structure on (010) plane.

**Figure 2 materials-15-07694-f002:**
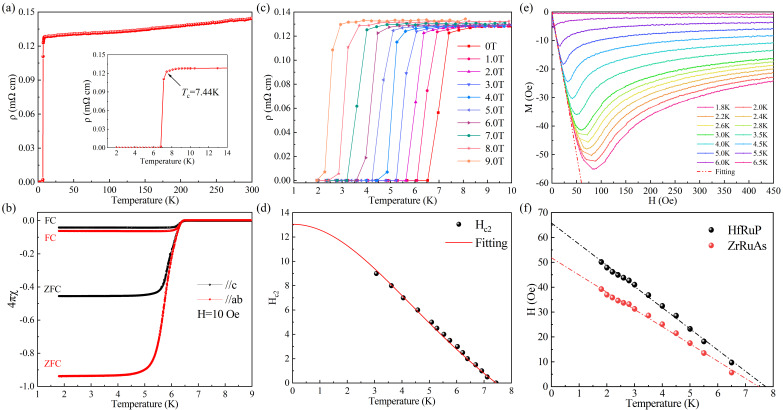
Electrical resistivity of ZrRuAs without (**a**) and with (**c**) field. The inset of (**a**) is the electrical resistivity of ZrRuAs lower than 14 K. (**b**) Temperature dependence of the magnetic susceptibility of ZrRuAs, measured in an applied field of 10 Oe using both the ZFC and FC protocols. (**d**) The upper critical field *H_c_*_2_ versus the transition temperature *T_c_* for ZrRuAs. The solid red line represents a fit to the G-L formula. (**e**) The field-dependent magnetization was recorded at various temperatures of ZrRuAs. For each temperature, *H_c_*_1_ was determined as the value where *M*(*H*) starts deviating from linearity (see red dashed line). (**f**) The lower critical field *H*_c1_ versus the transition temperature *T_c_* for HfRuP and ZrRuAs. The dashed line represents a linear fitting.

**Figure 3 materials-15-07694-f003:**
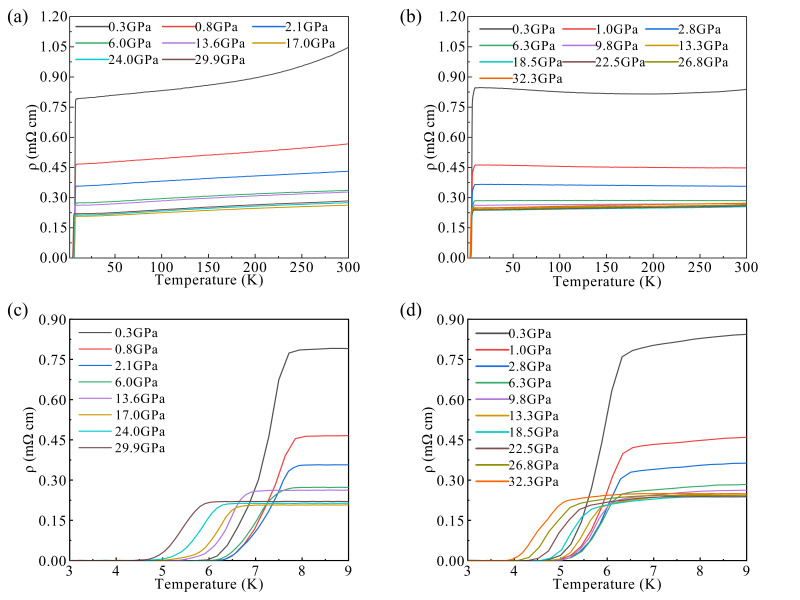
Electrical resistivity of ZrRuAs (**a**) and HfRuP (**b**) as a function of temperature at various pressures. Temperature-dependent resistivity of ZrRuAs (**c**) and HfRuP (**d**) in the vicinity of the superconducting transition.

**Figure 4 materials-15-07694-f004:**
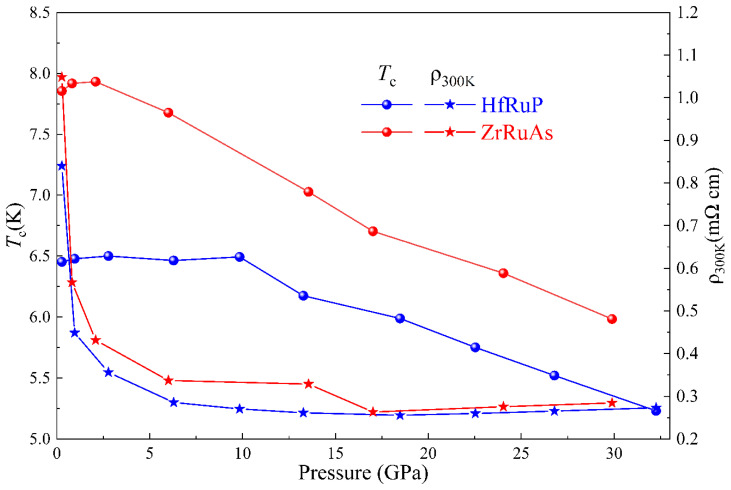
The pressure dependence of *T_c_*s and resistivity at 300 K of ZrRuAs and HfRuP.

**Figure 5 materials-15-07694-f005:**
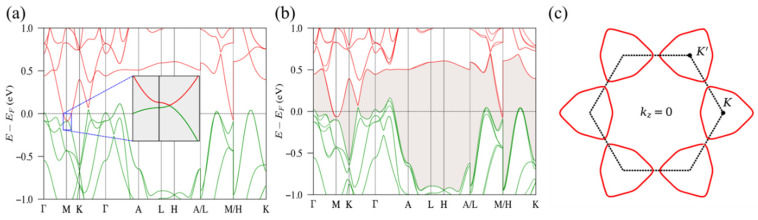
Electronic band structure and of ZrRuAs at 0.3 GPa without (**a**) and with (**b**) SOC. Green and red lines are valances and conduction bands. The inset in (**a**) shows a crossing along *M*-*K* in the absence of SOC. The shadow area in (**b**) indicates a continuous gap between valance and conduction bands when the SOC is turned on. (**c**) The nodal lines are presented on the *k_z_* = 0 plane when SOC is turned off. Red lines are nodal lines. Dashed lines indicate the first Brillouin Zone.

**Figure 6 materials-15-07694-f006:**
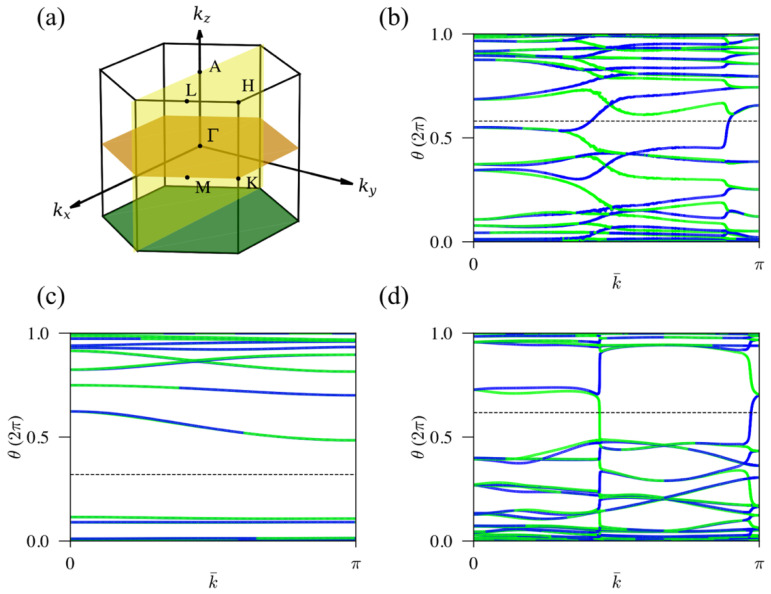
Mirror symmetric planes and Wilson loops at 0.3 GPa. (**a**) Orange, green, and yellow planes are mirror symmetric planes at *k_z_* = 0, *π*, and *k_y_* = 0, respectively. Wilson loops on the *k_z_* = 0, *π*, and *k_y_* = 0 are presented in (**b**–**d**), respectively. The horizontal dashed lines in each plot are the reference lines. Blue lines and green lines denote the flows of Wannier charge centers for states in subspaces of mirror *+i* and *−i* eigenvalues, respectively.

**Table 1 materials-15-07694-t001:** Mirror Chern number under different pressures. The first row denotes pressures in the unit of GPa. The second and third rows are mirror Chern numbers for *k_z_* = 0 and *k_y_* = 0 planes, respectively. Mirror Chern numbers for *k_z_* = *π* are always 0 under pressures in the table.

Pressure (GPa)	0.3	0.9	2.1	6.0	13.5	17.1	24.0	29.8
*C_M_* (*k_z_* = 0)	2	−2	2	−2	2	2	2	2
*C_M_* (*k_y_* = 0)	2	−2	−2	2	2	−2	−2	−2

## Data Availability

Not applicable.

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
