# Peer review of "Pressure-Tuning Superconductivity in Noncentrosymmetric Topological Materials ZrRuAs"

_materials, 2022, doi:10.3390/ma15217694_

Round 1
Reviewer 1 Report
In the presented work entitled „Pressure-tuning superconductivity in noncentrosymmetric topological materials ZrRuAs” by C. Li et al., the Authors investigate superconducting properties of the selected ternary transition metal pnictides. The analysis is performer by using the combination of the experimental and theoretical methods. The Authors show that the investigated materials are the type II superconductors which exhibit decreasing superconducting properties along with the increasing pressure.
In my opinion, the reported results may be of interest to the scientific community but unfortunately their presentation does not meet standards of the peer-reviewed journal. Therefore, I would like to reconsider the present paper after the following corrections:
1. I believe that usage of words such as “tremendous” (see Abstract) is an exaggeration. I also urge the Authors to use spaces between the text and reference i.e. instead of “(…) described elsewhere[8].” please write „(…) described elsewhere [8].”
2. In reference to the statement “The details are described elsewhere[8].” (see line 59) kindly provide brief description of the growth process, instead of just referring readers to the other study. In my opinion this is not a proper way of informing the readers about the details of any employed process/methodology. Similar critique is related to the sentence in lines 71 and 72.
3. The Authors give no motivation to investigate only the Fe2P-type ZrRuAs structure. In general the motivation to perform the analysis is relatively weak.
4. I believe that two expressions that include the critical field (see line 132 and 142) should be properly numbered. The Authors should also provide necessary references for less experienced readers in terms of the equation in line 132.
5. I see no purpose for the gradient background in Fig. 4.
6. I see no real motivation to investigate pressure-dependence of the critical temperature and resistivity in the discussed materials. Although the Authors refer to the high-pressure superconducting materials such as the selected hydrides, I see no essential relation between them and the ternary transition metal pnictides. The Authors does not provide nor refer to the reasoning analogous to the one made by Ashcroft for the hydrides. This is to say, their entire motivation is the fact that there are other superconducting materials where the phase of interest is influenced of even build on the pressure-like effects. Here the pressure is applied ad hoc without any rationale and for no obvious reason. I don’t know what the Authors would like to obtain within this process. In fact I cannot get any new and relevant insight based on the obtained result other than the negative influence of pressure on the superconducting properties in the analyzed ternary transition metal pnictides.
7. The discussion of the results obtained within the DFT calculations does not provide in my opinion any interesting findings. The Authors describe their result in details but do not inform the reader what about their results is actually interesting. Small related remark, the Authors refer to the DFT as the ab initio method but DFT is not a wave-function based method and in principle shouldn’t be called an ab initio method.
8. Is the electron-phonon mechanism responsible for superconductivity in the discussed materials? If yes why the Authors perform calculations without analyzing the electron-phonon properties of the materials of interest.
9. The conclusions are practically non-existent. In this section the Authors write just three sentences which are the extremely brief summary of the results described already in the previous sections. These are not conclusions by definition, just an extremely brief summary.
Reviewer 2 Report
The report is attached below

Round 2
Reviewer 1 Report
The Authors addressed most of my concerns, unfortunately there are still few aspect left without the proper answer. While one can argue whether or not DFT is truly an ab initio method (citing Wikipedia at the academic level shouldn't take place), I cannot agree that the motivation and conclusions got enough attention. The former is still weak in my opinion, the Authors made some changes to the text but the big picture is still missing. I urge the Authors to answer these questions in the text:
1. Why the Authors decide to investigate ZrRuAs under pressure?
2. What is the physical motivation to do the above?
3. What insight the Authors expect to gain based on the above (practically what is the hypothesis)?
As for the conclusions I still cannot accept them as a conclusions. The Authors added literally one sentence to the previous three sentence section. This is still not a conclusion. I will repeat myself, it is not enough to repeat the description of the results in the conclusions section, these are not conclusions by definition.
In the context of the above, the manuscript still does not meet formal requirements for the journal paper. I urge the Authors to address properly my above critique within the text of their manuscript. After that I am willing to reconsider my decision one more time.
